# Exploring the Role of Epithelial–Mesenchymal Transcriptional Factors Involved in Hematological Malignancy and Solid Tumors: A Systematic Review

**DOI:** 10.3390/cancers17030529

**Published:** 2025-02-05

**Authors:** Rimsha Kanwal, Jessica Elisabetta Esposito, Bilal Jawed, Syed Khuram Zakir, Riccardo Pulcini, Riccardo Martinotti, Matteo Botteghi, Francesco Gaudio, Stefano Martinotti, Elena Toniato

**Affiliations:** 1Centre of Advanced Studies and Technology, Department of Innovative Technology in Medicine and Dentistry, G.d’ Annunzio University, 66100 Chieti, Italy; kanwalrimsha809@gmail.com (R.K.); j.elisabetta.esposito@gmail.com (J.E.E.); bilaljawed2007@gmail.com (B.J.); khuramabbas512@gmail.com (S.K.Z.); riccardo.pulcini@unich.it (R.P.); e.toniato@unich.it (E.T.); 2Unit of Clinical Pathology and Microbiology, Miulli Generale Hospital, 70021 Acquaviva delle Fonti, Italy; 3Residency Program in Clinical Oncology, Faculty of Medicine, Umberto I University Hospital, University of Rome “La Sapienza”, 00185 Rome, Italy; riccardo.martinotti@uniroma1.it; 4Experimental Pathology Research Group, Department of Clinical and Molecular Sciences, Universita Politecnica delle Marche, 60126 Ancona, Italy; matteo.botteghi@worldconnex.com; 5Unit of Haematology, Department of Medicine and Surgeon, F. Miulli University Hospital, LUM University, Casamassima, 70010 Bari, Italy; 6Unit of Clinical Pathology, Department of Medicine and Surgeon, F. Miulli University Hospital, LUM University, Casamassima, 70010 Bari, Italy

**Keywords:** epithelial–mesenchymal transition, epithelial–mesenchymal transcriptional factors, hematological malignancy, solid tumor, cancer progression

## Abstract

In cancer research, the EMT is recognized as a hallmark of cancer metastasis, and a leading cause of cancer-related deaths. During the EMT, epithelial cells lose their cell–cell adhesion properties and acquire mesenchymal traits, enabling them to evade the surrounding tissues and migrate to distant sites. Several transcriptional factors are involved in the activation of the EMT. Our study focuses on how EMT transcriptional factors (e.g., SNAIL, TWIST, and ZEB) drive the EMT mechanism in both hematological malignancies and solid tumors. Understanding the role of EMT-TFs in cancer could lead to the development of targeted therapies to inhibit metastases.

## 1. Introduction

The epithelial–mesenchymal transition (EMT) is a dynamic process where epithelial cells shed their tightly connected characteristics and adopt mesenchymal properties. This transition is marked by reduced cell adhesion, the reorganization of the cytoskeleton, and increased cellular plasticity, enhancing the mobility and invasiveness [1] (Figure 1). The EMT plays a crucial role in both physiological and pathological processes, including embryonic development, inflammation, wound healing, fibrosis, and cancer progression [2]. The EMT is classified into the following three types based on its biological roles: Type 1, which drives embryogenesis, Type 2, which supports tissue regeneration, and Type 3, which promotes cancer progression [3]. Cancer represents a significant societal, public health, and economic challenge in the 21st century [4]. Globally, an estimated 19.96 million new cancer cases and 9.74 million cancer-related deaths occurred in 2022 [5]. In cancer cell dynamics, the EMT represents a continuous and dynamic spectrum of transition between epithelial and mesenchymal states. Tumor cells in the intermediate stages of this spectrum exhibit both epithelial and mesenchymal traits, enhancing their ability to survive, metastasize, and colonize distant organs [6]. The transcription factors such as SNAIL1, SNAIL2, TWIST1, TWIST2, ZEB1, and ZEB2 are the key drivers of the EMT. These EMT-TFs work in coordination, regulating each other’s activity and interacting with other transcription factors to influence target gene expression. They primarily function by suppressing epithelial genes (E-cadherin, Claudins, Cytokeratin, and Laminin 1) and promoting the activation of mesenchymal genes (N-cadherin, fibronectin, and vimentin), thus facilitating the transition [7]. EMT-TFs are activated by a range of growth factors (TGFβ, EGF, and VEGF), cytokines (IL6, IL10, and TNFα), signaling pathways (Wnt, Notch, P13K/AKT, and MAPK), and microenvironmental cues, collectively regulating the expression of genes that drive the epithelial-to-mesenchymal transition [8]. In the context of hematological malignancy and solid tumors, EMT-TFs are pivotal in regulating key aspects of cancer cell behavior, including tumor progression, metastasis, therapy resistance, immune evasion, and the acquisition of stem-like properties [9].

There is limited research that offers a comprehensive perspective on the role of EMT-TFs in both hematological malignancies and solid tumors, as much of the existing literature tends to focus on one type of cancer. This review aims to explore the role of EMT-TFs in both cancer categories, emphasizing the molecular mechanisms driving cancer progression and treatment resistance. By offering deeper insights into the significance of the EMT in cancer, this review highlights how a better understanding of these pathways may could lead to the development of therapeutic strategies to mitigate the detrimental effect of the EMT in cancer.

## 2. Materials and Methods

In July 2024, we conducted a systematic review to analyze the role of the EMT in hematological malignancies and solid tumors. The review encompassed English-language literature published between 2010 and 2024, with searches performed on PubMed and Google Scholar (Table 1).

Articles were filtered based on the following inclusion criteria: English language, publication within the past 14 years, full-text availability, and relevance to the EMT and EMT-TFs in hematological malignancies and solid tumors. Studies unrelated to these topics were excluded. Duplicate results were evaluated only one time. Abstracts, reviews, commentary, editorials, posters, and guidelines were excluded. Articles published in languages other than English were excluded. We evaluated the titles and abstracts and excluded the articles that were mismatched. Discrepancies were thoroughly discussed during the screening process. Every disputed case was carefully investigated and debated until a consensus was reached to ensure that high-quality and relevant studies were considered for the systematic review.

## 3. Results

The PRISMA flow diagram schematically depicts the selection process for the studies [10] (Figure 2).

The literature search identified 3250 publications. After removing 850 duplicates and screening the titles and abstracts, 1975 publications were excluded. A total of 425 articles were assessed, with 310 excluded for not meeting the inclusion criteria. A total of 115 publications were assessed for eligibility. We excluded studies in our final screening for the following three reasons: (1) they did not depict the role of the EMT in hematological malignancies or solid tumors; (2) they studied the EMT but not in the context of cancer; (3) they did not study the association between EMT-TFs and cancer. A total of 92 publications were finally selected for this systematic review.

### 3.1. Hematological Malignancy

Hematological malignancy or blood cancer affects the bone marrow, blood, or lymphatic system. These malignancies originate from the abnormal proliferation of blood cells. Of the cancers diagnosed in 2017 in the US, almost 10.2% were hematological malignancies [11]. Hematopoietic cells originating from mesoderm have a background of a mesenchymal developmental origin. All forms of hematological malignancies (lymphoid and myeloid leukemia, multiple myeloma, and lymphomas) show the expression of signatures, like the EMT, due to the uplifting of mesenchymal markers, such as vimentin, which are involved in aggressive tumor behavior [12,13]. The EMT is a reversible mechanism in which epithelial cells are swapped for mesenchymal cells, in which intercellular connections are disrupted, cell polarity is lost, the cytoskeleton is reorganized, and cellular movement is enhanced in cells that undergo the EMT process [14]. The transition of cells from the epithelial to the mesenchymal is driven by key transcription factors (TFs) [15]. These transcription factors suppress the genes associated with epithelial characteristics and promote the mesenchymal phenotype [16]. The major and master regulators of the EMT are the basic helix loop helix (BHLH) transcription factors TWIST1 and TWIST2, the zinc-finger-binding transcription factors SNAIL1 and SNAIL2, and the zinc finger E box-binding transcription factors ZEB1 and ZEB2 [8]. These transcriptional factors directly bind to the *E-cadherin* and suppress the *E-cadherin* gene. While other transcriptional factors such as E12/E47, Tbx3, FoxC2, SNAI3 (SLUG), and goosecoid can regulate EMT-related genes, their connection to *E-cadherin* expression is indirect and context-dependent, making them less central [17]. SNAI1, SNAI2, ZEB1, ZEB2, TWIST1, and TWIST2 are evolutionarily conserved and play similar roles in the EMT across various species. This makes them central players in understanding the universal molecular mechanisms of the EMT. These transcription factors have been extensively validated in functional studies, including genetic knockdown, and overexpression experiments, which demonstrate their necessity and sufficiency to induce the EMT [18]. EMT-TFs are indulged in pathological and physiological processes in living organisms such as cancer progression, wound healing, inflammation, fibrosis, and embryogenesis [2,19]. The EMT develops invasive, migratory, and metastatic properties in tumor cells [20]. The current literature strongly proved the association between EMT-TFs and the poor prognosis of patients [21,22,23,24,25,26,27,28] (Table 2)

#### 3.1.1. ZEB1

ZEB1, a prominent transcription factor of the EMT, plays a significant part in stemness and the progression of acute myeloid leukemia (AML). Stavropoulou et al. [21] reported that the higher expression of *ZEB1* is observed in AML patients with the hostile and stem-cell like phenotype. *Zeb1* knockdown via short hairpin RNA (shRNA) in an AML mouse model induced by the MLL-AF9 oncogene led to reduced infiltration in the bone marrow during an in vivo study and impaired tumor cell invasion in an in vitro study. This study demonstrated the role of *ZEB1* in leukemic cell dissemination. Stavropoulou et al. [21] further concluded that AML derived from long-term repopulating HSCs (LT-HSCs) had a higher expression of *ZEB1* as compared to AML derived from the granulocyte/macrophage progenitor, suggesting an association between *ZEB1*, stem-like, and immature AML. Furthermore, AML originating from LT-HSCs have higher counts of leukemia-initiating cells (LICs), which make the treatment and eradication of AML more complicated. Li et al. [29] further highlights the active role that *ZEB* plays in AML malignancy. They reported a higher level of *ZEB1* expression in AML patients as compared to healthy controls, emphasizing its role in AML progression and worse patient survival. The downregulation of *ZEB1* in human AML cell lines using short interfering RNA (siRNA) during in vitro study contributed to the expression of myeloid cell surface markers and the halting of cell proliferation, and eventually putting off the onset of tumors in xenograft models, suggesting its role in sustaining AML malignancy. This study also demonstrated a mechanistic association between the tumor suppressor pathway TP53 and the *ZEB1* level in AML cells. The downregulation of *ZEB1* elevated the TP53 protein levels and upregulated *ZEB1* downgraded the TP53 protein levels, indicating that *ZEB1* promoted the survival of cancer cells by suppressing this tumor suppressor pathway. Contrary to these findings, Almotiri et al. [30] found a lower expression of *ZEB1* in patients suffering from AML as compared to its healthy counterpart. The knockdown of *Zeb1* via Cre-mediated knockout in either a Hoxa9/Meisa1 or MLL-AF9 mouse model of AML in in vivo led to tumor progression, suggesting a complex role of *ZEB1.* Further, Bassani et al. [31] reported the association between *ZEB1* and immune modulation. *ZEB1* promoted the expansion of Th17 cells via upregulating factors (IL-23 and TGF-β), which secretes IL-17 cytokines in AML cells, thereby promoting tumor growth and immune suppression, thus contributing to AML progression. Sanchez-Tillo et al. [23] observed that ZEB1 also played a role in resistance to drugs in mantle cell lymphoma by switching on the genes involved in the proliferation (*MYC* and *MK167*), suppressing the proapoptotic (TP53 and BAX) and antiapoptotic (MCL1 and BCL2) pathways, and the membrane transporter proteins involved in drug efflux and influx (gemcitabine, doxorubicin, and cytarabine), thereby facilitating resistance against chemotherapy. They also demonstrated that Salinomycin is involved in the downregulation of *ZEB1* and inhibits the Wnt signaling pathway. This downregulation reduced the function of MDR1, leading to the promotion of drug retention within MCL cells and enhancing MCL cell sensitivity to chemotherapy drugs (gemcitabine, doxorubicin, and cytarabine). Targeting *ZEB1* might be a novel therapeutic approach to overcome drug resistance. Luanpitpong et al. [32] further described that a higher expression of *ZEB1* in MCL cells enhanced its resistance to Bortezomib and increased the growth potential of a lymphoma spheroid. Sun et al. [33] and Wu et al. [34] proposed *ZEB1* expression reduction in early T-cell precursor acute lymphoblastic leukemia (ETP-ALL). The *ZEB1* expression at the transcription level suppressed by the *LMO2* oncogene was particularly linked to the phenotypic characteristics of the ETP-ALL. A negative association existed between ZEB1 and *LMO2* in the ETP-ALL cells. LMO2 inhibited the ZEB1 DNA binding ability and reduced its activity. Wu et al. found that the reduced expression of *ZEB1* promoted the stemness phenotypic property in T-ALL and resistance against methotrexate therapy, which is used to treat T-ALL.

#### 3.1.2. ZEB2

ZEB2 is a key regulator of the EMT process, playing a critical role in the migration, invasiveness, and metastasis of hematological malignancy by repressing epithelial genes and promoting mesenchymal markers [35]. Further studies have suggested that ZEB2 exhibits an opposing role to ZEB1 in AML and ETP-ALL. Saia et al. [36] reported that *ZEB2* expression is not specifically elevated in AML cells, and that there is no correlation between the survival rate of AML patients and the *ZEB2* expression level. However contrasting results were presented by De-Conti et al. [37], who observed that *ZEB2* played an oncogenic function in the significance of promoting the progression and development of AML. Its expression was upregulated by the transduction of the *PML-RARα* and *AML1-ETO* oncogenes in the pre-leukemic hematopoietic stem cell. The knockdown of *ZEB2* led to the delay in leukemia progression. Further, Saia et al. [36] also observed that *Zeb2* expression significantly increased after the *AML-ETO* oncogene transduction into the hematopoietic progenitor cell line of mice. In addition to that, Shi et al. [38] observed a correlation between *ZEB2* expression and the upgrading expression of *ZEB2-AS1* long non-coding RNA, thereby promoting disease aggressiveness. Wang et al. [39] demonstrated that the knockdown of *Zeb2* via RosaERT2 Cre-mediated knockout in the mouse AML model derived from the MLL-AF9 oncogene led to the delay of leukemia progression. Interestingly, there was no additional delay in the progression of leukemia observed when both *Zeb1* and *Zeb2* from the MLL-AF9 model were knocked out. This confirms that *Zeb1* knockdown does not aid in the promotion of *Zeb2* in delaying leukemia progression, and that *ZEB2* has a specific role in the pathogenesis of AML that is independent of *ZEB1.* Strikingly, when either *Zeb2* or *Zeb1* were overexpressed in the hematopoietic system of mice, they significantly increased the extramedullary hematopoiesis development and myeloid compartment expansion, emphasizing their roles in myeloid cell proliferation. Similar explorations were performed by Li et al. [40] by using the CRISPR-mediated knockout or the shRNA-mediated knockdown of *ZEB2,* which induced myeloid differentiation and reduced the cell growth in the AML cell lines of humans in vitro. Furthermore, the shRNA-mediated knockdown of *Zeb2* in mice MLL-AF9 AML cells caused a reduction in the leukemia cell proliferation. These findings highlight the oncogenic role of *ZEB2* in AML progression. Further studies have found a higher *ZEB2* expression in early T-cell precursor acute lymphoblastic leukemia (ETP-ALL) patients as compared to acute myeloid leukemia patients. The higher expression of *ZEB2* negatively correlated with miR200C, a microRNA which is acknowledged to repress the *ZEB2* phenotype [28]. Gossens et al. [41] identified that a novel *BCL11B-ZEB2* fusion with its fusion partner 5′ BCL11B enhanced the *ZEB2* expression in ETP-ALL. The correlation between *ZEB2* and ETP-ALL was observed in hematopoietic transgenic mice after the development of ETP-ALL in the mice [42]. A previous study also discovered that *ZEB2* and LSD1 interaction in ETP-ALL transgenic mice enhanced the proliferation and stemness activity of ETP-ALL cells, making the treatment more difficult [43].

#### 3.1.3. TWIST1

TWIST1, a key EMT transcription factor, participates actively in the progression of various hematological malignancies. Myelodysplastic syndrome (MDS) is the clonal disorder of hematopoietic stem cells, which results in the abnormal differentiation and maturation of blood cells, and enhances the resistance against proapoptotic signals. There is a high risk in the advanced stages of MDS of developing into acute myeloid leukemia, which makes the treatment more difficult [44]. The higher expression of TWIST1 in the cells enhances the resistance against TNF-α-induced apoptosis, and promotes disease development and cell survival. *TWIST1* knockdown in MDS cells makes them more sensitive to TNFα-induced cell death by regulating the apoptotic process with the TP53, NF-kB, and miRs10a/b interaction, showing an association between resistance mechanisms and *TWIST1* [45]. MDS patients that resist the treatment of DNA demethylating agent 5-aza-2-deoxycytidine had a higher expression of TWIST1 as compared to responsive patients, further highlighting its role in therapeutic resistance [46]. Wang et al. [47] further demonstrated that the enforced expression of TWIST1 in KG1a (an AML cell line) increased the sensitivity to the chemotherapy drug cytarabine, but there was no modification observed in the daunorubicin response. Contrary to this, another discovery indicated that the enforced overexpression of TWIST1 in AML cell lines (K562 and U937) enhanced the resistance activity to imatinib, mitoxantrone, and daunorubicin; the study further found that a higher expression of TWIST1 was linked with a worse overall survival in the AML samples. These contradictory findings determined that the role of TWIST1 in AML may depend on specific drug combinations or genetic contexts. Furthermore, Cosset et al. [27] studied the association between tyrosine kinase inhibitor (TSI) treatment and TWIST1 expression in chronic myeloid leukemia (CML). The expression of TWIST1 was 100X greater in the CML patient samples that did not respond to the TKI treatment as compared to those that did respond. The TWIST1 expression was lower in the imatinib-sensitive CML cell line in contrast to the resistant CML cell line. Moreover, the expression of TWIST1 has not yet been studied in B-cell or T-cell acute lymphoblastic leukemia (B-All/T-ALL), but it is significantly expressed in ALK+ anaplastic large cell lymphoma (ALCL) and cutaneous T-cell lymphoma (CTCL) [24]. Goswami et al. [48] described that the TWIST1 expression level in CTCL was associated with the progression of the disease level from the indolent mycosis fungoides stage to the aggressive Sezary syndrome stage. The expression of TWIST1 increased in the Sezary syndrome stage due to either the acquired promoter hypomethylation or the chromosomal region 7p21, which stresses its part in disease development and severity. Furthermore, Che-Chen et al. [49] emphasized the oncogenic role of TWIST1. It prevents c-Myc-induced apoptosis by antagonizing the p53 pathway, which plays a critical role in cell death and tumor suppression. Additionally, TWIST interacts with the NF-ĸB pathway, protecting cells from TNF-α-induced apoptosis and sustaining the oncogenic activity. Chang et al. [50] demonstrated that the NF-ĸB pathway is constitutively active in Sezary syndrome and is responsible for increased cell survival and proliferation. Michael et al. [51] suggested that using a combination of four genes, including *TWIST1,* could lead to the accurate and early diagnosis of Sezary syndrome. Zhang et al. [26] studied the TWIST1 expression in ALCL, a pediatric lymphoma impelled by the fusion of t (2;5) NPM-ALK. The knockdown of *TWIST1* from the cell lines of ALK+ ALCL increased their sensitivity to Crizotinib, a tyrosine kinase inhibitor, and reduced the invasiveness. This study also confirmed that TWIST1 contributes to disease prognosis and therapeutic resistance. The *TWIST1* implication has been studied in Multiple Myeloma (MM), particularly in ~15% patients with MM with the (4;14) translocation. This translocation ameliorated the expression of the NSD2 oncogene in MM patients, which has a correlation with EMT gene signatures. The *TWIST1* knockout from the NSD2+ MM cell lines reduced the invasiveness and EMT gene signature downregulation in in vitro studies. Conversely, the enforced expression of TWIST1 increased its dissemination in vivo and migration in vitro in a mouse MM cell line, but did not exert its influence on proliferation and tumor growth [28].

#### 3.1.4. TWIST2

TWIST1 and TWIST2 exhibit opposing roles. TWIST2 has tumor-suppressive functions, which is in contrast to the oncogenic role of TWIST1. The hypermethylation of TWIST2 leads to its loss in AML [52]. Zhang et al. [52] emphasized the tumor-suppressive nature of TWIST2 in AML. *TWIST2* knockout in AML cells enhanced the ability to form colonies and to grow faster, indicating that *TWIST2* loss enables the cells to overcome critical growth constraints. The enforced expression of TWIST2 in AML cells reduced the proliferation of cells and the generation of new colonies. Mechanistically, the expression ofTWIST2 was found to activate the *CDKN1A* gene (cell cycle regulator) and repress the tumor-progressive genes, indicating its tumor-suppressive nature. Similarly, the hypermethylation of *TWIST2* genes has also been observed in both adult and childhood ALLs, involving both T-cell and B-cell lineages [53]. Thathia et al. [53] further observed that the treatment of cells with 2′-deoxy-5 azacytidine, a DNA demethylating agent, restored the TWIST2 expression and enhanced the progression and development of leukemia. These studies showed that TWIST1 participates actively in disease progression, metastatic malignancy, poor prognosis, and therapeutic resistance in hematological malignancy. In contrast, TWIST2 functions as a tumor suppressor, and its loss is correlated with a higher malignancy potential. Understanding the discrete functions of TWIST2 and TWIST1 is imperative in the advancement of therapies to eradicate drug resistance and to improve the outcomes in cancer.

#### 3.1.5. SNAI1

SNAIL is a vital transcription factor of the EMT, playing a vital role in hematological malignancy. Like TWIST and ZEB, SNAIL plays a role in AML progression [54,55,56], as well as drug resistance [55] and poor prognosis [56]. Carmichael et al. [56] explored the oncogenic role of SNAI1 in AML progression by developing hematopoietic-restricted SNAI1 transgenic mice. These mice showed a myeloproliferative phenotype that eventually transitioned to AML after a considerable latency stage. The mice expressed skewed macrophage/granulocyte lineage progression, with a higher number of immature myeloid cells with a slightly impaired differentiation and increased self-renewal capabilities. Carmichael et al. [56] studied the physical interaction between SNAI1 and the histone lysine demethylase LSD1, an interaction that disrupts LSD1 function and its role in epigenetic regulation. This interaction disrupted the DNA binding patterns and led to the abnormal regulation of targeted genes related to inflammatory pathways, cellular adhesion/invasion/migration, cytokine signaling, and normal myeloid differentiation. Jiang et al. [57] investigated the role of SNAIL in the histone deacetylase inhibitor (HDACI)-induced EMT. HDACIs are a kind of anticancer drug that significantly participate in the progression of hematological malignancy via the upregulation of *SNAIL*. HDACI treatment led to lower levels of E-cadherin and the uplifting of vimentin and SNAIL, a transition which is the hallmark of the EMT. HDACIs enhance the SNAIL acetylation and decreases its ubiquitylation, which enhances the SNAIL activity and stability. HDACIs also promote its nuclear translocation and transcription. The *SNAIL* knockout via siRNA reduced the upregulation of Vimentin and morphological changes.

#### 3.1.6. SNAI2

SNAI2, a crucial transcriptional factorassociated with the EMT, is also significantly enhanced in AML, and plays a vital part in the pathogenesis of AML. The AML oncogenes *HOXA9, MEIX1,* and *MLL-AF9* were enhanced the SNAI2 expression via viral transduction in HSCs. The knockdown of *Snai2* reduced the AML oncogene (*NUP98-HOXA9* and *MLL-AF9*) ability to alter the mouse HSCs, which showed that SNAI2 is important for the transformation of oncogene and AML progression. The reduced frequencies of leukemia-initiating cells (LICs) and leukemia stem cells (LSCs) in MLL-AF9 further highlights its role in sustaining leukemogenesis. *Snai2*-deficient MLL-AF9 AML cells showed normal homing; nevertheless, abnormal cell cycle progression and increased apoptosis was observed. It plays a prominent part in cell proliferation and survival [58]. Zhang et al. [58] further confirmed these results by the knockdown of *SNAI2* in human AML cell lines, resulting in the reduction in LIC/LSC frequencies and decreases in the proliferative capacity. Further, Mancini et al. [59] highlighted the role of SNAI2 in CML and its association to chemotherapy resistance. The *BCR-ABL* oncogene directly drives the overexpression of SNAI2 in CML. *BCR-ABL* is a fusion oncogenethat promotes cell proliferation and survival in CML. TKI is used for the treatment of *BCR-ABL* and halts its oncogenic activity. Mancini et al. found that TKI treatment in CML cells reversed the overexpression of SNAI2, which promoted the discharge of the proapoptotic protein called PUMA. The PUMA restoration induced apoptosis in CML cells, which showed that, under oncogenic influence, SNAI2 repressed the cell death mechanisms. The higher expression of SNAI2 was observed in TKI-resistant *BCR-ABL* mutation CML samples when compared to TKI-sensitive samples. This enhanced expression sustained cell survival and promoted drug resistance in resistant cases. Given the similarity between AML and CML in terms of oncogene-driven mechanisms, it is likely that the overexpression of SNAI2 in AML plays a similar role in promoting resistance and cell proliferation to standard AML therapies. These investigations highlight that both are key regulators of therapy resistance and leukemia progression.

**Table 2 cancers-17-00529-t002:** The role of the EMT in hematological malignancy.

Transcriptional Factor	Hematological Malignancy	Role/Function	Reference
ZEB1	Acute Myeloid Leukemia (AML)	Promotes the EMT; increases the stemness and aggressiveness of the disease Regulates the self-renewal ability of leukemic stem cells; enhances the AML malignancy; drug resistance and poor prognosis	[21,29,31]
	Mantle Cell Lymphoma (MCL)	Reduces apoptosis; enhances proliferation; chemotherapy resistance and worse overall survival	[23,32]
	Acute Lymphoblastic Leukemia (ALL)	Promotes stemness and therapy resistance	[33,34]
ZEB2	Acute Myeloid Leukemia (AML)	Oncogenic activity promotes leukemia stemness, proliferation, and progression	[36,37,39]
	Acute Lymphoblastic Leukemia (ALL)	Enhances the proliferation and stemness activity of cells; promotes drug resistance	[43]
TWIST1	Myelodysplastic Syndrome (MDS)	Expression is linked to disease development, reduced apoptosis, enhance cell survival, and therapeutic resistance	[44,45,46]
	Acute Lymphoblastic Leukemia (ALL)	Resistance to imatinib, mitoxantrone, and daunorubicin; enhances cell proliferation and increases apoptosis, which is associated with worse overall survival	[47]
	Chronic Myeloid Leukemia (CML)	Expression level is 100X greater than AM associated with tyrosine kinase inhibitor (TKI)	[27]
	Cutaneous T-cell Lymphoma (CTCL)	Expression is associated with disease progression from mycosis fungoides to Sezary syndrome	[48,49,50,51]
	Anaplastic Large Cell Lymphoma (ALCL)	Promotes invasiveness and chemoresistance	[26]
	Multiple Myeloma (MM)	Enhances invasiveness and metastasis	[28]
TWIST2	Acute Myeloid Leukemia (AML)	Tumor suppressor; reduces colony formation and cell growth	[52]
	Acute Lymphoblastic Leukemia (ALL)	Tumor suppressor; enhances chemosensitivity and reduces proliferation	[53]
SNAI1	Acute Myeloid Leukemia (AML)	Promotes drug resistance, impaired differentiation, and self-renewal	[56]
SNAI2	Acute Myeloid Leukemia (AML)	Enhances proliferation and disease progression; inhibits apoptosis, which is crucial for maintaining the LIC and LSC population	[58]
	Chronic Myeloid Leukemia (CML)	Promotes chemotherapy resistance and cell survival	[59]

### 3.2. Solid Tumors

Solid tumor malignancy is indeed strongly dependent on the adjacent stromal cells, i.e., the immune cells, mesenchymal stem cells (MSCs), and cancer-associated fibroblasts (CAFs) [60]. These stromal cells secrete inflammatory cytokines and growth factors, such as TGF-β and activin A, which trigger the EMT pathways and stimulate the migration of cancer cells through the MEK/ERK and P13K/AKT pathways [61,62]. Solid tumors have become a critical health challenge globally. In 2022, lung cancer was the most diagnosed cancer, accounting for 2,480,301 cases (12.4%), followed by female breast cancer, with 2,295,686 cases (11.5%), prostate cancer, with 1,466,680 cases (7.3%), and gastric cancer, with 968,350 cases (4.9%). Lung cancer was also the leading cause of cancer-related deaths, responsible for 1,817,172 deaths (18.7%), followed by liver cancer (757,948 deaths, 7.8%), female breast cancer (665,684 deaths, 6.9%), and gastric cancer (659,853 deaths, 6.8%) [5].

EMT-TFs have a critical role in the migration, invasion, and metastasis cascades of solid cancers [63] (Figure 3 and Table 3). The dual nature of genes makes the understanding and treatment of cancer more complex. In some studies, PROX1 operates as a tumor suppressor, with the reduced migration and invasion of breast cancer cells [64]. Another study discussed the oncogenic role of PROX1. According to this study, PROX1 interacts with hnNRPK to trigger the Wnt/β-catenine pathway, which directly activates the EMT process. The EMT process ameliorates the metastatic colonization and invasion of breast cancer cells [65]. NLRP3 inflammasome is a main player in enhancing breast cancer cell metastasis and invasion. NLRP3 does so by enhancing IL-1β level, which, in turn, mediates the EMT and develops a pro-inflammatory environment that promotes tumor progression [66].

Similarly, in lung cancer, cytokines have a significant impact on cancer prognosis, such as the upregulation of IL-17A in the lung tumor-induced overexpression of NLRP3 and the enhanced lung tumor invasiveness ability mediated by the EMT [67]. TGF-β is a key regulator of the EMT in NSCLC, which promotes the EMT and the increased invasiveness and aggressiveness of lung cancer [68]. The prostate is an androgen-dependent organ, and the androgen/AR signaling pathway plays a vital role in the organogenesis and tumorigenesis of the prostate. The androgen receptor (AR) not only controls the organogenesis and tumorigenesis of the prostate, but also increases metastasis by promoting signaling interactions, as well as through the EMT [69]. TGF-β induces the EMT in prostate cancer in vitro via the suppression of E-cadherin levels, and elevates the expression of fibronectin, vimentin [70], and EMT-TFs such as ZEB1 and SNAI1/2 [71]. A higher level of IL-6 was observed in the metastatic specimens of prostate patients [72], and plays an important role in EMT induction, leading to prostate cancer invasiveness [73]. SLUG is a key driver of prostate cancer metastasis via metalloprotease secretion, cytokine production, and EMT induction. SLUG suppresses the *KISS1* (metastasis suppressor gene) and enhances the metastasis of prostate cancer. Clinically, *KISS1* loss is highly observed in metastatic and primary prostate cancer as compared to localized ones. *KISS1* expression restoration in metastatic prostate cancer cell lines reduces cell invasion motility [74]. Similarly, the overexpression of ZEB1 [75] and TWIST [76] was observed in malignant prostatic cancer as compared to benign tumors. In ovarian cancer, Lysophosphatidic Acid (LPA) promotes metastasis and progression. This promotion is linked to the downregulation of the *SIRT1* gene (EMT suppressor and ZEB1 inactivator) [77] and the upregulation of SLUG/SNAI2 [78]. The ovaries are responsible for the secretion of various hormones and growth factors, such as activin A and TGF-β, which trigger EMT pathways and stimulate the migration of cancer cells through the MEK/ERK and P13K/AKT pathways [79]. In pancreatic cancer, ZEB1 upregulation is a key factor in promoting metastasis. A key factor in this process is Musashi 2 (MSI2), which enhances the epidermal growth factor (EGF) expression that activates the EGF receptor (EGFR), leading to ZEB1 upregulation. ZEB1 stimulates the ERK/MAPK signaling pathway, which further enhances pancreatic cancer invasion [80]. The metastasis of pancreatic cancer is associated with colon cancer protein 1 (MACC1), which positively interacts with SNAI1; this interaction represses *CDH1* (which encodes E-cadherin) and enhances fibronectin 1 (FN1) expression, which promotes cell migration. Together, these effects enhance the metastatic and invasive ability of pancreatic cells by promoting the EMT [81]. *Helicobacter pylori* (*Hp*) infection can induce the EMT in stomach cells. *Hp* may promote the EMT by enhancing MMP-7 levels, which, in turn, increases the HB-EGF level [82]. *Hp* cytotoxin-associated gene A (Cag A) reduced the expression of E-cadherin and enhanced the expression of vimentin and TWIST1, thereby describing new EMT signaling pathways in gastric cancer cell lines [83]. Norepinephrine (NE) induced the EMT process in gastric cancer cells through β2-AR-HIF-1α-SNAIL activity, reduced E-cadherin expression, and enhanced the EMT markers, such as vimentin, which further promoted the invasiveness and migration of gastric cancer cells [84]. There is a correlation between the EMT phenotype and advanced gastric cancer stages [85]. Gastric cancer cells in the early stage shows a non-EMT phenotype, indicating a lower tendency to metastasize and with reduced migration. In contrast, gastric cancer cells in the advanced stage exhibit the EMT phenotype induced by Fas signaling, indicating higher metastasis and migration in the cells [86]. SNAIL is the prominent inducer of hepatocellular carcinoma (HCC), which increases the stemness capability of cancer cells and chemotherapeutic resistance; therefore, the combination of TIP30 with E-cadherin can serve as a potent biomarker for the prognosis of HCC [87]. The Wnt/β-catenin pathway is the key driver of the EMT in HCC patients. Catenins are expressed in the cell membrane and sometimes the nucleus; the mutation in its gene leads to the diffusion of E-cadherin, which is implicated in intrahepatic lesions and satellite nodules. The mutation in the *CTNNB1* gene, which is responsible for β-catenin nuclear translocation, is an indicator of a progressive EMT [88]. Yu et al. [89] investigated how TWIST and SLUG are upregulated in bladder cancer cells, whereas SNAIL expression is downregulated in bladder cancer cells, and TWIST expression is positively correlated with the tumor progression, grade, and stage, while the downregulation of E-cadherin is associated with advanced tumor grades and stages of bladder cancer. Collectively, the upregulation of TWIST and SLUG, and the downregulation of SNAIL and E-cadherin, are associated with poor survival and more aggressive phenotypes of bladder cancer.

EMT-TFs also play a significant role in anticancer drug resistance, which makes the treatment difficult [90]. The study on breast cancer evaluated that the overly expressed EMT-TFs, such as FOXC2, SNAIL, and TWIST, enhanced the promoter activity of ATP-binding cassette (ABC) transporters. This association between the upregulation of ABC transporters and EMT-TFs proposes that the EMT is the key critical regulatory factor in chemotherapy resistance [91] (Figure 4).

Rosan et al. [92] studied the part of endothelin-1 (ET-1) which is involved in the development of the EMT phenotype and chemotherapy resistance in ovarian cancer. They used the Taxol- and cisplatin-resistant EOC cell model in vitro and in vivo, and observed that ET-1 activated *SNAIL* and enhanced the cancer development. Researchers found Notch-2 activation in pancreatic cancer cells, which experienced the EMT process and showed resistance against gemcitabine [93]. Gemcitabine disrupts the transcription process and synthesis of DNA, and is used for pancreatic cancer treatment. The EMT promotes resistance against gemcitabine through several mechanisms. The EMT causes epigenetic changes and reduces the henT1 expression, a nucleoside transporter that is crucial for gemcitabine uptake; moreover, the EMT overexpressed ABC transporters such as MRP1 and MDR1, which actively efflux the gemcitabine from pancreatic cancer cells, thus decreasing their effectiveness [94]. Paclitaxel is considered as a primary chemotherapeutic drug for the treatment of gastric cancer. Its efficacy was reduced due to the EMT. Several studies observed that the expression of the EMT in parental gastric cancer cells indicates the resistance to paclitaxel [95]. The overexpression of SLUG and SNAIL with the epidermal growth factor receptor (EGFR) mutation in lung cancer cell lines induced the resistance against gefitinib [96]. The PAX6-ZEB2 axis enhanced cisplatin resistance and metastasis in NSCLC via the P13K/AKT signaling pathway [97]. Hypoxic condition also induced EMT in HCC, as a result invasion, metastasis and drug resistance increased in HCC. The P13/AKT/HIF-1α pathway in hypoxia enhanced the EMT, which led to worse treatment outcomes [98]. SNAIL promoted AR activity and led to the resistance to AR-targeted therapies like enzalutamide. It was significantly overexpressed in metastatic tumor cells as compared to benign tumors [99]. LSD1 (lysine-specific demethylase 1) has emerged as a significant player in promoting chemoresistance, proliferation, and metastasis. The higher expression of LSD1 was observed in bladder cancer tissues, particularly in those patients who had undergone chemotherapy. The upregulation of LSD1 in bladder cancer was correlated with poor prognosis, metastasis, and advanced cancer grades. Functional studies showed that the knocking down of *LSD1* inhibited the EMT process and reduced the cancer progression [100]. The role of EMT-TFs in drug resistance and metastasis make them attractive therapeutic targets in solid tumors.

**Table 3 cancers-17-00529-t003:** Role of the EMT in solid tumors.

Solid Tumors	Role of EMT	Reference
Breast Cancer	NLRP3 inflammasome promotes the IL-1β level that, in turn, stimulates the EMT; as a result, this increases the tumor progression EMT-TFs increased the drug resistance via enhancing the promoter activity of ABC transporters	[66,90]
Ovarian Cancer	The ovaries secrete hormones and growth factors, such as activin A and TGF-β, which induce the EMT and stimulate the migration of cancer cells through the MEK/ERK and P13K/AKT pathways ET-1 activates *SNAIL* in Taxol- and cisplatin-resistant EOC cell models	[79,92]
Pancreatic Cancer	The EGF receptor (EGFR) leads to the upregulation of ZEB1; ZEB1 stimulates the ERK/MAPK signaling pathway, which further enhances the PC invasion Notch-2 activation in pancreatic cancer cells that have undergone the EMT mechanism show resistance to gemcitabine	[80,93]
Gastric Cancer	*Helicobacter pylori* (*Hp*) infection promotes the EMT by enhancing MMP-7 levels, which, in turn, increase the HB-EGF level. *Hp* cytotoxin-associated gene A (Cag A) reduces the E-cadherin expression and enhances the vimentin and TWIST1 expression in gastric cancer cell lines The EMT induces resistance against paclitaxel in gastric cancer cell lines	[82,83,95]
Lung Cancer	The overexpression of SLUG and SNAIL in lung cancer cell lines with epidermal growth factor receptor (EGFR) mutation induces resistance against gefitinib The PAX6-ZEB2 axis promotes cisplatin resistance and metastasis in NSCLC via the P13K/AKT signaling pathway	[96,97]
Hepatocellular Carcinoma	The hypoxic condition induce the EMT in HCC; as a result, invasion, metastasis, and drug resistance increases in HCC. The P13/AKT/HIF-1α pathway plays an active role in the hypoxia-induced EMT, which leads to worse treatment outcomes	[98]
Prostate Cancer	SNAIL promotes AR activity and leads to the resistance to AR-targeted therapies like enzalutamide. SNAIL is significantly overexpressed in metastatic tumor cells as compared to benign tumors	[99]
Bladder Cancer	The upregulation of TWIST and SLUG, and the downregulation of SNAIL and E-cadherin, are associated with poor survival and more aggressive phenotypes of bladder cancer	[89]

## 4. Discussion

The epithelial–mesenchymal transition is a cornerstone of cancer development, progression, and metastasis, enabling carcinoma cells to acquire invasive, migratory, and stem-cell-like properties that promote metastasis. The EMT-TFs in solid tumors transit epithelial cells into mesenchymal cells, and this transition promotes invasion, migration, and metastasis [101]. Cancer-associated fibroblasts and macrophages within the tumor stroma release several growth factors that induce the EMT, such as TGFβ1, PDGF, EGF, and HGF [102]. Cytokines, including TNF-α, IL-10, IL-8, and IL-6, are also released by the tumor stroma, and are identified as chemotactic, immunosuppressive, and pro-survival signals, further supporting EMT induction [103]. In hematological cancers such as leukemia and lymphoma, EMT-TFs play a significant role by driving chemoresistance, immune evasion, stemness, and cellular plasticity rather than motility [104]. Deshmukh et al. [105] employed single-cell RNA (sc-RNA) sequencing to demonstrate the dynamic nature of the EMT. The researchers calculated the “EMT score” and observed that the EMT score increased over time. However, some of the cells showed a hybrid state even after 8 days. This heterogeneity complicates therapeutic targeting and highlights the adaptability of cancer cells. Our study shows that EMT-TFs have an association with poor survival, prognosis, and therapy resistance. The higher expression level of vimentin was positively correlated with white blood cell counts and worse overall survival in AML patients, as well as in solid cancers, which was also connected with worse patient outcomes [106,107]. Imani et al. [108] demonstrated that the expression of SNAI1 and TWIST1 had a correlation with the advanced stages of metastatic breast cancer and the worse overall survival of patients. Similarly, the study by Wan et al. [109] showed an association between the expression of EMT-TFs and potential poor prognostic factors in hepatocellular carcinoma patients. The study by Li et al. [29] demonstrated that a high ZEB1 expression was linked with the worse overall survival of AML patients. Additionally, a hypoxic environment developing in growing tumors before vascularization is another important factor that induces and enhances the EMT in cancer cells [110]. Hypoxia triggers several pathways for the stimulation of the EMT through the stabilization of HIF-1α [111]. HIF-1α stimulates EMT-TF activity and enhances cancer progression and chemotherapy resistance [112]. Zhang et al. [113] studied the resistant mechanisms in minimal residual disease and relapse in B-ALL. They observed that the leukemic cells at relapse shifted to a poorly differentiated state in contrast to a diagnostic state, while residual cells show more complex changes and diverse survival strategies as compared to the relapse state. The in vivo and in vitro experimental models showed that inhibiting the hypoxia pathway made the leukemia cells more sensitive to chemotherapy. This study suggests that targeting the hypoxia pathway could help to eliminate MRD cells and reduce the risk of relapse. Anand et al. [114] studied the resistant mechanism in relapsed/refractory ETP-ALL patients carrying the *NOTCH1* mutation by performing sc-RNA sequencing. The mutation in *NOTCH1* abnormally activated the notch signaling pathway. They identified two different stem-cell-like states that were distinct in terms of the oncogenic signaling and cell cycle. Fast cycling cells divide rapidly and show a higher activation of the notch signaling pathway, as well as sensitivity to notch inhibitors. Slow-cycling cells divide slowly and rely on P13K signaling for survival, and show resistance to notch-targeted therapies. Slow cycling develops an immunosuppressive environment and accumulates dysfunctional CD8+ T-cells to evade the immune response. This study suggests that a combination therapy targeting both the P13K and notch pathways could be more effective in the treatment of ETP-ALL. However, targeted EMT-TFs and signaling pathways could be beneficial in the treatment of cancer, but it has adverse effects and significant challenges. EMT-targeted therapies at the early stage of cancer prevent the proliferation, migration, and invasiveness of tumor cells, and make it easier to remove the primary tumor after surgery. In the advanced stages of the diseases, EMT inhibitors prevent the CTC colonization of the primary tumor, decrease the proportion of the CSCs in tumors, and reduce the CTC ability to develop secondary tumors at distant sites [115]. It is unclear whether EMT therapy is beneficial at the early and advanced stages, because the treatment showed a contrary result than originally expected. The reverse process MET, driven by the anti-EMT treatment, promoted the metastasis and colonization of the CTC and increased the proliferation [116]. The targeted process of the EMT not only controls the tumor cells but also has a negative impact on normal cells [117].

## 5. Conclusions

The EMT plays a pivotal role in cancer progression and development, with EMT-TFs being extensively implicated in promoting chemoresistance, antiapoptotic mechanisms, stemness, metastasis, invasion, and cellular proliferation. The elevated expression levels of EMT-TFs are closely associated with poor therapeutic responses and overall survival. Chemoresistance remains a major challenge in cancer treatment, with some anticancer drugs inadvertently enhancing EMT-TF expression, thus further exacerbating tumor progression. These findings highlight the potential of EMT-TFs as therapeutic targets. Future research should focus on developing targeted drugs and novel therapeutic strategies to mitigate EMT-related cancer progression and to improve treatment outcomes.

## Figures and Tables

**Figure 1 cancers-17-00529-f001:**
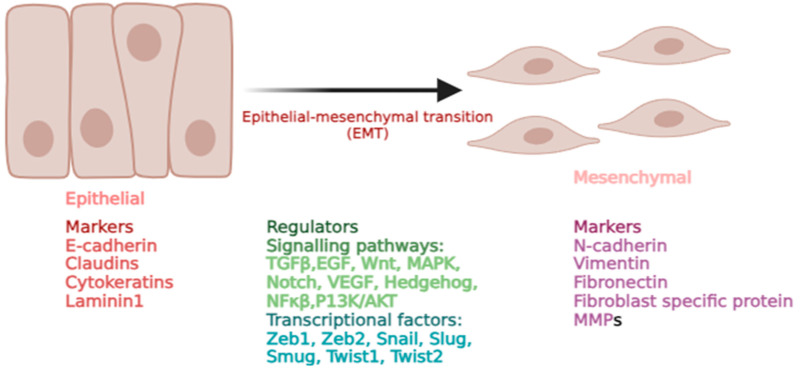
The epithelial–mesenchymal transition (EMT).

**Figure 2 cancers-17-00529-f002:**
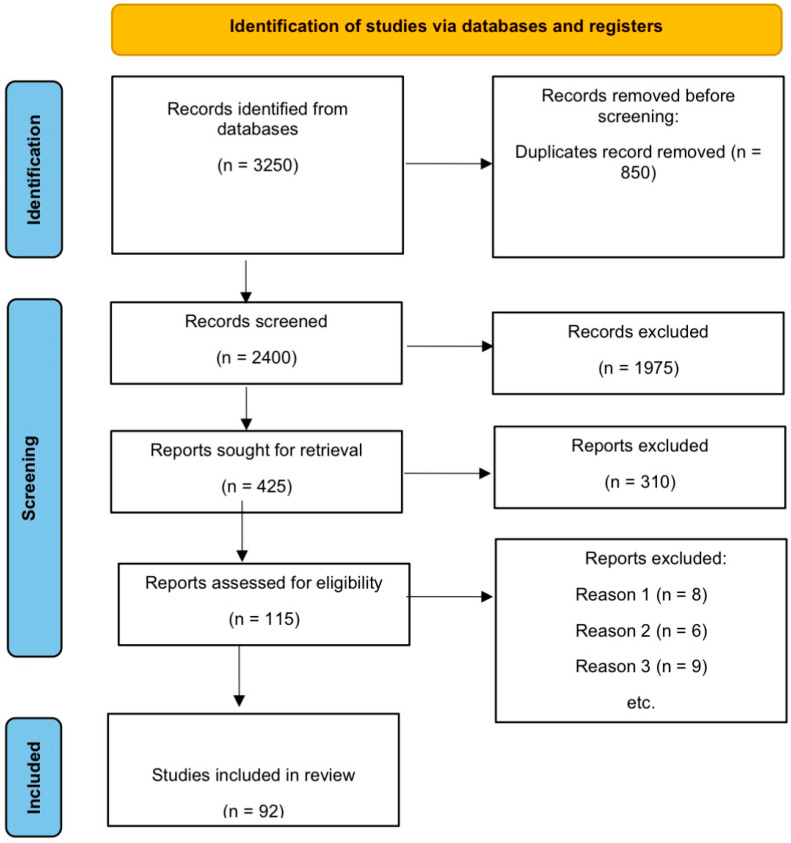
Prisma flow diagram.

**Figure 3 cancers-17-00529-f003:**
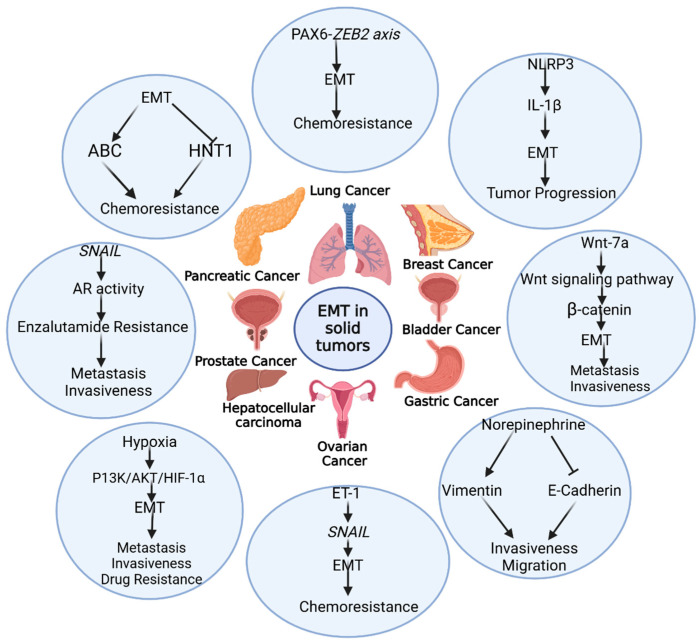
Role of the EMT in solid tumors.

**Figure 4 cancers-17-00529-f004:**
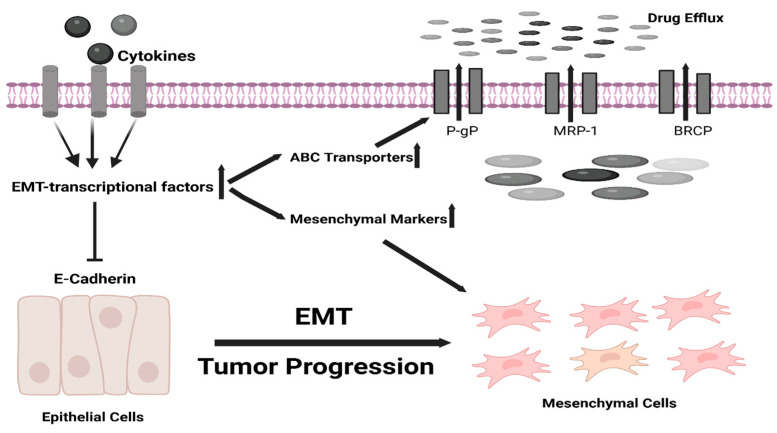
EMT and ABC transporters.

**Table 1 cancers-17-00529-t001:** Keywords used for the literature search.

Database	Keywords
PubMed	(“Epithelial Mesenchymal Transition” [tiab] OR “EMT transcriptional factors” [tiab] OR “EMT-TFs” [tiab]) AND (“haematological malignancy” [tiab] OR “haematopoietic tumour” [tiab] OR “solid tumours” [tiab] OR “tumorigenesis” [tiab] OR “carcinogenesis” [tiab]) NOT (“review” [pt] OR “systematic review” [pt])
Google Scholar	Epithelial Mesenchymal transition OR EMT OR EMT transcription factors OR EMT-TFs OR EMT and haematological malignancy OR EMT-TFs and haematological malignancy OR EMT and solid tumors OR EMT-TFs and solid tumors

## Data Availability

All data supporting the findings of this study are available within the article and from the corresponding author upon reasonable request.

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
