# Peer review of "Exploring the Role of Epithelial–Mesenchymal Transcriptional Factors Involved in Hematological Malignancy and Solid Tumors: A Systematic Review"

_cancers, 2025, doi:10.3390/cancers17030529_

Round 1
Reviewer 1 Report
Comments and Suggestions for Authors
In this manuscript, Kanwal R et al. summarized the recent studies about EMT-related transcription factors from PubMed and Google Scholar. Manuscript is well written, but several points to be improved.
1; They do not give a clear definition of “EMT-TFs”. TF seems to be an abbreviation for “transcriptional factor”, but they should specify this in manuscript. Since there are many type of transcription factors existed, they should first define EMT-TFs as being limited to Zeb1, Zeb2, Snail, Slug, Smug(?), Twist1, and Twist2.
2: They did not describe the structures of some transcription factors in EMT-TFs. I think this should be stated. ZEB1 and ZEB2 belong to the zinc-finger E-box binding homeobox (ZEB) family, Snail1 and Snail2 (Slug) belong to the C2H2-type zinc finger transcription family.
3: In line 356, “TWIST1 and TWIST2 both are the members of basic helix-loop-helix (bHLH) of EMT transcription factors, but exhibit opposing role.” May be better to move to line 312.
4: In Figure 1, they show “Smug” as transcriptional factor, but did not mention in this manuscript. What is this ? If this is not necessary for this review, please delete it.
5: There are also transcription factors that regulate these EMT-TFs in expressional or functional levels. For example, nuclear receptors are one of the major transcription factor families which regulate gene expression by binding to lipid-soluble small molecules, and there are many reports of NRs regulating EMT via regulation of EMT-TFs expression, etc. I searched PubMed as “EMT transcriptional factors. “nuclear receptor””, and found 34 articles from 2010 to 2024. Among them, the following article seems to show a link to EMT-TFs, but they do not cite. I would like to know the reason why they did not mention them. If they felt necessary for citing these papers, please write and cite them.
Retinoid orphan nuclear receptor alpha (RORalpha) suppresses the epithelial-mesenchymal transition (EMT) by directly repressing Snail transcription.
Xiong G, Xu R. J Biol Chem. 2022 Jul;298(7):102059.
Elevation of androgen receptor promotes prostate cancer metastasis by induction of epithelial-mesenchymal transition and reduction of KAT5.
Lin CY, Jan YJ, Kuo LK, Wang BJ, Huo C, Jiang SS, Chen SC, Kuo YY, Chang CR, Chuu CP. Cancer Sci. 2018 Nov;109(11):3564-3574.
NR2F1 Is a Barrier to Dissemination of Early-Stage Breast Cancer Cells.
Rodriguez-Tirado C, Kale N, Carlini MJ, Shrivastava N, Rodrigues AA, Khalil BD, Bravo-Cordero JJ, Hong Y, Alexander M, Ji J, Behbod F, Sosa MS. Cancer Res. 2022 Jun 15;82(12):2313-2326.
High NR2F2 transcript level is associated with increased survival and its expression inhibits TGF-beta-dependent epithelial-mesenchymal transition in breast cancer.
Zhang C, Han Y, Huang H, Qu L, Shou C. Breast Cancer Res Treat. 2014 Sep;147(2):265-81.
Reviewer 2 Report
Comments and Suggestions for Authors
The submitted work is a review article on epithelial-mesenchymal transition in hematological and solid tumors, which is a broad and general topic.
The systematic review provided valuable data, which is a significant work and is appreciated.
Detailed Comments:
Abstract: The Abstract correctly depicts the outcome of the systematic review, clearly describes that from 3250 found studies, which main outcomes regarding EMT and treatments came out. According to this Abstract, the provided knowledge is not a minimal further to a stand several years ago. So, the Abstract suggests that EMT is related with tumor progression, poor outcomes and chemotherapy resistance and if nothing is to mention about an EMT-targeting therapeutic approach, then, although several tousand papers were published in the last 10 years, but there is no progress.
Introduction
The Introduction contains information, which is fully available in previous review articles. This does not make sense to repeat this material again.
Material and Methods
The systematic review method was appropriate.
Results
Due to the correctly performed systematic review, the Results provide a valuable data bank on tumor types and molecules in relation to EMT.
Discussion
The provided information in this EMT review is not much new. Based on the correct method used, it looks like, that this image is characteristic for the related literature, as no real breakthrough in the literature since more than 15 years. EMT concerns a small cell population usually recognized by single cell sequencing. The past few years provided RNASeq profiles of single cell sequencing on EMT cell populations, which might contain tumor type-specific and general information. This contribution to the literature might be significant or not, but it is worth to discuss. It is also interesting, how to try to develop therapy against a low-represented cell population, which will later cause recurrence and residual tumors? When should the EMT cells, which are not really correctly detected by classical diagnostic approaches, be eliminated? Before surgical tumor removal, systematic therapy or perform therapy and after that try to eliminate the EMT cells or the tumor growing from them after MET? How the research helps the clinical challenges? As you can see from this evaluation, a more critical approach would serve the readers.
Reviewer 3 Report
Comments and Suggestions for Authors
This is a semi-review depicting the role of EMT transcriptional factors in leukemia. This is an important issue that has not been subjected to exhaustive review, thus it is a valuable entry in the catalog of such papers. The authors have performed systematic meta-analysis of the literature to define, or predict, specific roles for SNAIL, SLUG, TWIST1/2 and ZEB1/2 in different types of leukemia. A couple of issues need to be addressed, but this is an adequate contribution.
In the introduction, the authors mention the role of microRNA in EMT. It would be useful to add a Table with the most important miRNA involved in EMT.
While important for the development of the analysis, the part of the text on the role of EMT transcriptional activators in solid tumors is highly redundant and should be condensed or removed.
The main issue is that EMT is likely to be important in the formation of solid lymphoma, e.g. Sezary syndrome, where hematopoietic cells “have reason” to interact with their microenvironment. This seems to be a major target of the modifications seen in conventional EMT, thus this needs to be explored in more detail. Since the authors have probably exhausted the existing knowledge in this regard, the authors are more than welcome to speculate, particularly addressing the genes regulated by EMT TFs that would provide leukocytes in lymphoma with an environmental advantage.
Also, the connection between EMT and development of resistance to diverse types of therapy has been established in solid tumors. This could be explored in the context of leukemia in more detail here, at least from a theoretical perspective.
Round 2
Reviewer 2 Report
Comments and Suggestions for Authors
I did not consider before that your focus is on hematological malignancy. All the answer to the comments are appropriate.